# WiGId: Indoor Group Identification with CSI-Based Random Forest

**DOI:** 10.3390/s20164607

**Published:** 2020-08-17

**Authors:** Xiaochao Dang, Yuan Cao, Zhanjun Hao, Yang Liu

**Affiliations:** 1College of Computer Science and Engineering, Northwest Normal University, Lanzhou 730070, China; dangxc@nwnu.edu.cn (X.D.); cy18017320479@outlook.com (Y.C.); liuyang_nwnu@126.com (Y.L.); 2Gansu Province Internet of Things Engineering Research Center, Lanzhou 730070, China

**Keywords:** human identification, wireless sensing, channel state information, random forest

## Abstract

Human identity recognition has a wide range of application scenarios and a large number of application requirements. In recent years, the technology of collecting human biometrics through sensors for identification has become mature, but this kind of method needs additional equipment as assistance, which cannot be well applied to some scenarios. Using Wi-Fi for identity recognition has many advantages, such as no additional equipment as assistance, not affected by temperature, humidity, weather, light, and so on, so it has become a hot topic of research. The methods of individual identity recognition have been more mature; for example, gait information can be extracted as features. However, it is difficult to identify small-scale (2–5) group personnel at one time, and the tasks of fingerprint storage and classification are complex. In order to solve this problem, this paper proposed a method of using the random forest as a fingerprint database classifier. The method is divided into two stages: the offline stage trains the random forest classifier through the collected training data set. In the online phase, the real-time data collected are input into the classifier to get the results. When extracting channel state information (CSI) features, multiple people are regarded as a whole to reduce the difficulty of feature selection. The use of random forest classifier in classification can give full play to the advantages of random forest, which can deal with a large number of multi-dimensional data and is easy to generalize. Experiments showed that WiGId has good recognition performance in both LOS (line of sight) and N LOS (None line of sight) environments.

## 1. Introduction

With the increasing popularity of wireless devices, more and more people begin to study the use of wireless devices for indoor location, human behavior perception, and human identification. Traditional identification methods are mainly realized by some auxiliary devices, such as wearing ID cards to represent identity, biometric identifiers, such as fingerprints, iris, and so on. ID cards require users to wear special cards, while fingerprint recognition requires users to have contact with the machine, which has a certain risk of spreading viruses. The iris and facial recognition methods take the facial features of the human body as features and identify them with the help of cameras. It has high accuracy, but it is easily affected by the lighting in the environment, and there are privacy problems, so it can only be used in a specific environment.

However, the contactless identification method using Wi-Fi devices does not require users to wear additional devices, does not require human-to-human contact, and is not affected by external natural environments, such as temperature, humidity, and light, so it has a wider application prospect. The channel state information (CSI) is more reliable than the RSSIs that change frequently depending on the time and environment [1], and CSI can provide more information than RSSI; therefore, more and more scholars mainly study the technology of human perception and recognition based on CSI. Gait information can be used as a feature of identification [2]. The researchers ask testers to walk between Wi-Fi devices and identify each individual by extracting gait features; when pedestrians walk through Wi-Fi devices, in addition to gait information, other body features, such as waving and height and weight, also affect the received data, but these are not regarded as features. Therefore, there are some defects. Wi-Who [3] proposed a framework that uses Wi-Fi devices to identify an individual from a group of people. The gait of the human body is collected as a feature, and the individual to be identified can be identified from a group of two to six people. WifiU [4] extracts fine-grained gait information and uses gait information as a feature to distinguish different people. WFID [5] is a kind of passive free indoor human identification system that uses a new feature based on frequency, establishes the relationship between people and the proposed new feature, and classifies it by support vector machine (SVM), which has high classification accuracy. Zou [6] used fine-grained gait information as fingerprint features to design a new Internet of things application platform, which uses machine learning algorithms for classification and can get 91% recognition accuracy in a group of 20 people. Lv [7] used the principal component analysis (PCA) algorithm to process the CSI data collected from the Wi-Fi equipment, selected the most important features from the gait information that can be extracted, used the improved SVM to classify, and obtained high recognition accuracy with a low computational cost. Reference [8] proposed a personal identification method based on gesture features, which uniquely identifies users by establishing the relationship between different user gesture features and CSI. It has high accuracy in the environment where the multipath effect is simple, but it can hardly play a role in the environment where the multipath effect is complex. Reference [9] proposed a method that collects gait information of users and then classifies and identifies them through neural networks. This method uses 23-layer deep convolution neural networks and can achieve an accuracy of about 90% from a group of 24 people at most. In reference [10], a method of identity recognition using the random forest as a classifier is proposed, which achieves a maximum accuracy of 97.5% in the indoor environment. To sum up, gait information is often regarded as the main feature of identity recognition. However, when pedestrians walk between Wi-Fi devices, the rest of the body information, such as height, weight, arm, and hand swing, will affect the signal transmission, so researchers’ identification should not only take gait as a feature but should study the overall characteristics of people. Wang [11] studied human fall movements, and the interest of the research is at the health care level of the elderly. Wi-Fi equipment is arranged in the living environment, which can be reminded when falling movements are detected. The method proposed in this paper can achieve a detection accuracy of up to 91%. Zhang [12] paid attention to the detection of a fall, which is a special human movement, and put forward a method to detect the fall of the human body in real-time. Wang [13] modeled the human body, put forward the concept of Fresnel zone, and perceived the human respiratory rate on the basis of this concept, and realized the accurate perception of fine-grained movements. Based on the theory of the Fresnel zone, the position, height, and direction of the equipment needed to detect breathing are discussed. Dang [14] adopted the data processing method of the combination of PCA and Kalman filter and used the idea of a fingerprint database to study human behavior detection in different environments. The movements detected include squatting, falling, and turning, etc., and the people are taken as a whole to extract features. The accuracy of detection is 93%. Reference [15] used the method of deep learning to identify the movements of the human body and distinguish different people, using the idea of collecting a large number of data, and then putting the data into a model with strong computing power for training. A method of indoor human behavior detection is proposed in [16]; in addition to the amplitude information in CSI, it is studied that the phase information of CSI is affected by human activities, and the proposed phase feature has a certain ability to resist environmental interference, so this method has certain robustness and can be applied to many environments. Li [17] proposed a new crowd counting method. By combining the amplitude information and phase information of CSI data, the robustness of the method is improved, and the performance of detection is also improved. Li [18] studied how to use commercial Wi-Fi equipment to detect people without a pre-training classification model, added speed detection as the basis of human activities, and achieved the highest detection accuracy of 100%. References [19,20,21,22] focused on the methods of detecting people’s movements, discussed the influence of personnel movements on signal propagation, and studied how to identify people’s movements with high accuracy in a changeable environment. In particular, Zhu [21] studied the receiving angle of the signal, discussed the influence of the signal angle on the action of the recognition person, and proposed a comprehensive phase and amplitude information to locate and identify the movement of the human body. In references [23,24], the authors deeply discussed CSI, and put forward a method of detecting human motion based on CSI, established speed and motion model, and verified the performance of the method in different experimental environments. Wu [25] proposed a human perception model based on CSI. Cao [26] mainly applied the method of machine learning to the classification of human behavior and compared the performance of different machine learning methods. Most of the methods that use Wi-Fi to perceive people are often environment-dependent. One method has high accuracy in a specific experimental environment, but in other environments, the accuracy of the method may be greatly reduced. Reference [27] proposed a deep-learning-based device-free activity recognition framework that can remove the environment and subject-specific information contained in the activity data and extract environment/subject-independent features shared by the data collected on different subjects under different environments. Reference [28] proposed an indoor localization scheme using signal strength that can be easily implemented in a smartphone, and the experimental results showed that the method has good performance in both line of sight (LOS) and none line of sight (NLOS) environments. Reference [29] proposed recurrent neuron networks (RNNs) for a fingerprinting indoor localization using Wi-Fi; this method is accurate, but it may be more accurate if CSI information is used. Reference [30] studied selected literature examples and even succeeded in devising fully untrained model-based solutions. Reference [31] presented a deep learning-based indoor localization system that achieves a fine-grained and robust accuracy in the presence of noise, and it has been proved that the system has good accuracy in different environments. Reference [32] proposed a hybrid of support vector machine (SVM) and deep neural network (DNN) to develop scalable and accurate positioning in Wi-Fi-based indoor and outdoor environments, and the experiment showed that this method could provide scalable positioning, and 100% of the estimation accuracies are with errors less than 1 m and 1.9 m for indoor and outdoor positioning, respectively. Reference [33] presented a human activity recognition system, which neither requires user instrumentation, nor specialized infrastructure, nor active operation; this system has been built into USRP software radios.

On the basis of previous studies, we consider that when studying human identification, we should not only take gait information as a feature but also take into account the height, shape, and elbow swing of the human body. In addition, in some special scenarios, the number of people going in and out may not be one at a time, such as four workers carrying heavy things to enter a laboratory with access control. At this time, it is very inefficient to identify each worker one by one, so it is meaningful to be able to identify a group of people at the same time. When there are a large number of people to be identified at the same time, some methods of extracting amplitude and phase from CSI as features are faced with great difficulty in feature extraction and data processing. One of the advantages of random forest is that it can deal with a large number of high-dimensional data; therefore, this paper presented a group identification method based on random forest fingerprint database. The method includes two stages. First, the CSI data is collected in the offline stage, then the original collected CSI data are processed by the noise removal method, and the preprocessed CSI data are used as random forest training data. After training the random forest, in the online stage, the real-time collected data can be input into the random forest after denoising, and the classification results can be obtained.

The contributions of this paper are summarized as follows:In the data preprocessing stage, this paper used a combination of PCA filtering and low-pass filtering, which can effectively remove the noise of CSI data and retain effective features. Subsequent experiments also showed that this method is effective.Using the random forest as a classifier, in the experimental environment of this paper, it is verified that the method based on random forest fingerprint has certain accuracy in identity recognition and compared with other algorithms.The performance of the proposed method is verified in three different environments, and the setting of the experimental environment takes into account the complexity of the multipath effect (the multipath effect in the laboratory is more complex than that in the open hall). The effects of LOS and NLOS environments on the performance of the method are also considered. The rest of the paper is organized as follows: Section 2 describes the preliminary. Section 3 describes in detail how to design the system. Section 4 introduces the experimental environment and analyzes the performance of this method through experiments and compares it with other methods. Finally, we have concluded the work in Section 5.

## 2. Preliminary

In this chapter, we have first introduced the CSI data and analyzed the impact of the number of people and human body characteristics on CSI; because there is often noise interference in the original CSI data, so we have further introduced the methods of CSI data preprocessing.

### 2.1. CSI Data Analysis

With the use of the 802.11/n protocol, we could extract CSI data from standard commercial Wi-Fi devices. CSI reflects the attenuation of Wi-Fi signals in the process of transmission. For signals transmitted using orthogonal frequency division multiplexing (OFDM) modulation technology, the signals received by the receiver can be expressed as:(1)$$\vec{Y}=H\vec{X}+\vec{N}$$
where $\vec{Y}$ and $\vec{X}$ represent the receiving and sending signal vectors, respectively, H represents the channel information matrix, and $\vec{N}$ represents the additive white Gaussian noise. The CSI of each subcarrier can be estimated from $\vec{X}$ and $\vec{X}$ as:(2)$$H=\frac{\vec{Y}}{\vec{X}}$$
where H varies, and the size of the matrix varies according to the hardware. In the hardware equipment using the Inter 5300 network card, the number of subcarriers is 30, and therefore, H can be expressed as *H* = [*H*_1_, *H*_2_, *H*_3_, …, *H*_30_]. CSI data can be expressed as a complex matrix of m×n×k, where m and n represent the number of antennas at the sender and receiver, respectively, and k represents the number of subcarriers. In this paper, Intel 5300 Network card was used to extract CSI data, so k is 30. The collected data is shown in Figure 1.

Figure 1a,b represent the CSI images of two different people walking between Wi-Fi devices, which can be analyzed. Different people have a significant impact on CSI, but the main reasons for this effect should be: (1) physiological characteristics of the human body, such as height, weight, (2) walking posture, speed, hand-shaking mode, and frequency, and other factors.

Because there is not a good model to study each individual in the concurrent perception, this paper studied the multi-person walking as a whole, which is equivalent to building a fingerprint database for the whole. With regard to the idea of building a fingerprint database for the whole, the subsequent experiments in this paper have discussed in detail the accuracy of detection. At the same time, through experimental verification, we regarded multiple people as a whole to study the upper limit of Wi-Fi perception based on the idea of the fingerprint database.

### 2.2. Data Preprocessing

In the process of CSI data acquisition, due to the influence of the equipment itself and the interference of the indoor complex experimental environment, there will be a lot of interference and noise in the collected data. There is a need to eliminate the noise in the original data in order to obtain clean and effective data for feature extraction and classification. Because the CSI data has a large amount of data and belongs to high-dimensional data, the principal component analysis (PCA) algorithm can be used to retain the most important components in the data and remove the secondary components. In addition, most of the movements of the human body occur in the low-frequency range; for example, the frequency of walking behavior analysis is about 20–80 Hz. Therefore, in order to extract more accurate features, the original data can be processed by a low-pass filter. In this paper, the preprocessing of the original CSI data mainly included two parts. First, the original CSI data was reduced by PCA, and the number of subcarriers changed from 30 to 1. Then, the high-frequency part of the original data was filtered by a low-pass filter. The data processing diagram is shown in Figure 2.

We can see from the figure that the processed CSI data is free from the interference of noise and retains the effective features we need. To be clear, the set of features is computed on the channel matrix H, which is preprocessed by low-pass filter and PCA filter. In order to retain as many identity features as possible, so as to make the method more accurate, all the preprocessed channel matrix H is used for training. At the same time, the use of the random forest as a classifier also takes into account the advantages of random forest in dealing with high-dimensional and complex data.

## 3. WiGId Method

This part introduces in detail the group identification method based on random forest fingerprinting, which has been carried out in detail from three parts: decision tree, random forest, and recognition method. The first section of this chapter introduces the construction of decision tree, the second section introduces the construction of random forest, and the third section introduces the two stages of WiGId—offline training phase and online matching phase.

### 3.1. Decision Tree

The decision tree is a tree-shaped structure, in which each internal node represents a judgment on an attribute, each branch represents an output of a judgment result, and each leaf node represents a classification result. The process of getting the classification result by the method of the decision tree is the process of starting from the top root node and going down to the leaf node step by step. The root node of the decision tree contains a complete set of samples, and each non-leaf node represents a segmentation of the samples. The decision tree used in the proposed method is a binary partitioning tree. Each non-leaf node is divided into two descendants, one on the left and one on the right. The segmentation algorithm of nodes is the key to the construction of a decision tree. Before introducing the segmentation algorithm of nodes, the basic theories that need to be introduced are introduced.

Information entropy: Information entropy can be simply understood as the uncertainty of information, which was first given by Shannon, and the mathematical formula for calculating information entropy is given, assuming that there are k samples in the current sample set D.
(3)$$Entropy(D)=-\sum_{i=1}^{k}\left(p_{i}\log_{2}p_{i}\right)$$
(4)$$p_{i}=\frac{C_{i}}{\left|D\right|}(i=1,2,\cdots \cdots ,k)$$

Equation (3) is the definition of information entropy, where |D| represents the number of samples in the sample set D, i represents the number of the sample set, $C_{i}$ represents the number of $i_{th}$ samples, and $p_{i}$ represents the proportion of $i_{th}$ samples to the sample set. According to the formula, it can be analyzed that the higher the information entropy, the lower the purity of the sample.

The decision tree needed is a tree in which the entropy of a sample decreases fastest, and the purity of nodes is getting higher and higher based on information entropy. Therefore, there is a need to measure each feature of the sample and select the best partition attribute as the split node. The commonly used measurement methods are information gain (ID3), information gain rate (C4.5), and Gini coefficient (cart).

The difference of information entropy before and after node splitting is called information gain, which can be expressed as:(5)$$Gain(D,A)=Entropy(D)-\sum {}_{i\in Values(A)}\frac{\left|D_{i}\right|}{\left|D\right|}Entropy(D_{i})$$
where D is the total input space, and $D_{i}$ is the subset of D for which attribute A has a value i. Generally speaking, the greater the information gain, the faster the decrease of information entropy, and the higher the purity of the branch sample set. So, the optimal partition attribute can be chosen as follows:(6)$$a_{*}=\arg\max Gain(D,A_{i})$$
where $A_{i}$ is the $i_{th}$ attribute used to divide the sample set D. It is proved that the segmentation algorithm with information gain as a measure tends to select attributes with more branches, and sometimes over-fitting occurs. Therefore, in order to reduce the bad influence caused by the preference of information gain, the information gain rate can be used to select the optimal partition attribute, and the information gain rate is defined as:(7)$$Gain\_ratio(D,A)=\frac{Gain(D,A)}{SI(D,A)}$$

The numerator is the information gain when splitting the node with the attribute A, and the denominator is called the split information measure of the attribute A, which is defined as:(8)$$SI(D,A)=-\sum_{i=1}^{D}\frac{\left|D_{i}\right|}{\left|D\right|}\log_{2}\frac{\left|D_{i}\right|}{\left|D\right|}$$
where $D_{i}$ is all the subsample sets obtained by splitting the sample set D according to attribute A.

In the process of constructing the decision tree, taking the information gain rate as the split standard when constructing the decision tree, the information gain rate can solve the problem caused by the information gain preference for attributes with many branches to a certain extent. Algorithm 1 introduces the pseudo-code of constructing the decision tree, collects the input data set D through the Intel 5300 network card, and trains the decision tree (DT) as the output.


**Algorithm 1.** Training Decision Tree
**Input: dataset**
D

**Output: the decision tree**
1. Initialize an empty tree2. Generate processed training dataset $D_{train}$3. repeat4. For each attribute a∈D5. Compute the gain_ratio of a6. End7. choose the best split attribute $a_{sp}$ based above computed criteria8. create a decision node and attach this node to the corresponding branch of the tree T9. partition the dataset to subdatasets based on $a_{sp}$10. for each subdatasets $D_{i}$11. Repeat same operation from 3 to 12.12. End13. until $D_{i}$ is pure or size of $D_{i}$ less than minimum or the algorithm reaches enough iterations14. return T.


In Algorithm 1, the original data is first processed by PCA and low-pass filter, and the processed data is used as the training data set of the decision tree. The preprocessing reduces the influence of noise in the data on the classification so as to improve accuracy. $a_{sp}$ is the best split attribute calculated according to the information gain rate. Step13 gives the termination condition of the algorithm, which has three different situations: 1. The sub data set has been pure. This means that there is no need for the data set (node) to be divided or split. 2. The number of samples in the node is less than the predetermined threshold minimum. 3. The decision tree has reached enough iterations. Generally speaking, the ordinary decision tree needs post-pruning to reduce the probability of over-fitting, but in the method proposed in this paper, the last classification task is random forest. The random forest has the ability to avoid over-fitting because of its inherent characteristics, so in algorithm 1, the post-pruning strategy is not adopted for the generated decision tree. The next section describes the random forest algorithm in detail.

### 3.2. Random Forest

The idea of integrated learning is to solve the inherent defects of a single model so as to integrate multiple models, learn from each other, and get the best results. Random forest is the integration of multiple decision trees, which can not only get a good classification accuracy but also has a strong generalization ability. The WiGId method proposed in this paper is to integrate the DT mentioned in the previous section, in order to achieve better classification results.

The core idea of random forest is bootstrap aggregating, a single data set can only generate one tree with the same algorithm, and the bagging strategy can help us generate different data sets. First of all, the original sample set D is preprocessed to remove noise to get the training sample set $D_{train}$, and the subsample training set $D_{train}^{i}(i=1,2,\cdots s)$, is obtained from the original sample set $D_{train}$ (sample size is s). Therefore, the training set for each tree is different. But it will contain repeated training samples. Secondly, if the feature dimension of each sample is M, then a constant m (m<<M) is set to randomly select m feature subsets $A_{m}$ from M features, and in the process of decision tree construction, the features based on each split are selected from $A_{m}$. Thirdly, in the process of building a random forest, each tree should grow to the maximum extent and do not need pruning. Finally, n decision trees are combined under the thought of a voting algorithm.

It should be noted that because of the two operations of a random selection of sample sets and random selection of features, random forests are not easy to fall into quasi-merging and have strong robustness. The construction code of random forests is given below:
**Algorithm 2.** Construction of Random Forest**Input: Originally collected CSI data packet**D**, each data packet contains**j**data for**N test cases**Input: the size of the forest: s****Output: random forest: F**1. generate training dataset $D_{train}$ by Wavelet transform2. for i = 1 to s do3. Generate new training dataset $D_{train}^{i}(i=1,2,\cdots s)$ by bootstrap aggregating4. set $m=\log_{2}M$5. randomly select m attributes from $D_{train}^{i}$6. use $D_{train}^{i}$ train the $i_{th}$
DT based Algorithm17. end8. combine the s Decision Trees on the basic thought of voting method.9. return F


The above code Algorithm 2 is the core of the WiGId proposed in this paper. Because the final classification result of random forest is decided by voting according to the results of multiple decision trees, it has higher accuracy than a single decision tree. The following is a flow chart for helping understand the whole construction process more clearly and intuitively.

Figure 3 intuitively shows the training process of the random forest. The final output of the random forest is based on the idea of voting and synthesizes the results of each decision tree.

### 3.3. The Framework of WiGId

The method we proposed includes two stages. In the offline stage, the Intel5300 network card is first used to collect CSI data, then the original CSI data D is simply filtered to remove part of the noise to get the processed CSI data $D_{train}$, and then the dataset $D_{train}^{i}$ used to train the random forest is obtained by bootstrap aggregating. On this basis, the random forest is constructed, and the real-time CSI data collected in the online phase are preprocessed by the same process. Then, it is input into the trained random forest, and finally, the estimated result is obtained. The flow chart is shown in Figure 4.

To sum up, the whole method can be divided into three main steps, the first step is to take the CSI data from the hardware, the second step is to preprocess the collected data into a data set that can be used to train the random forest, and then train the random forest, and the third step is to input the real-time data collected in the online phase into the random forest to get the results of classification and estimation.

## 4. Experiment and Analysis

This chapter discusses the performance of WiGId from the following aspects: (1) first analyze the performance of the proposed method in three different environments and then compare it with the previous algorithm of other paper (WFID [5], SVM (Radial Basis Function kernel) [7]) in three different environments; (2) discuss the influence of the depth of the decision tree and the size of the random forest on the accuracy of the method; (3) discuss the influence of the height of the equipment and the signal and packet sending rate on the accuracy of the method in the experimental environment; (4) study the maximum number of people that can be identified by WiGId.

### 4.1. Experimental Environment

Because the CSI information is easily affected by the external environment, the same method may get very different results in different environments, so the experimental environment of this paper is introduced in detail here. The experimental equipment includes two Lenovo desktop computers, which are used as transmitters and receivers. The operating system of the two computers is Ubuntu16.04LTS, and the CPU models of both computers are Intel Corei3-4150. The Intel5300 network card is used to extract CSI data. Because we wanted to study the influence of the NLOS environment on the detection of the method, the Wi-Fi frequency chosen is 2.4 GHz. The codes of the transmitter and the receiver are modulated, which can adjust different packet sending rates. There are two experimental sites, one is the wireless sensing laboratory of the Internet of things Center in Gansu Province, and the other is a relatively empty hall. The laboratory is equipped with tables and benches needed for experiments, and the multipath effect is complex and close to the place of daily life. Only the sender and receiver of CSI data collection are placed in the empty hall, and the external influence is relatively small. The area of the laboratory is 50 m^2^, and the area of the empty hall is 40 m^2^. The laboratory field map and layout are shown in Figure 5.

When collecting CSI data, the sender has only one antenna to send data, and the receiver has three antennas to receive data, so the CSI data received is a matrix of 1×3×30, that is, each group of data packets includes three sub-links, and each sub-link is composed of 30 sub-carriers. A total of 31 situations were set in the experiment, and there were five subjects in the experiment. According to the increasing number of people, they were divided into 5 cases of the one-person group, 10 cases of the two-person group, 10 cases of the three-person group, 5 cases of the four-person group, and 1 situation of the five-person group. In each experiment, all the subjects walk in their own habit along the direction perpendicular to the Fresnel area, and the walking distance is 4.5 m, and the duration is 5 s, so under the same duration, different packet frequency may have an impact on the effect of the experiment, and different equipment height may have different effects on the recognition results.

### 4.2. Accuracy Standard

Because we used the random forest to solve the classification problem, the most common criteria for judging the classification results are the true positive rate (TP rate), false positive rate (FP rate), and accuracy. They are defined as follows:
(9)$$TP\;Rate=\frac{True\;Positive}{True\;Positive+False\;Negative}$$
(10)$$FP\;Rate=\frac{False\;Positive}{True\;Negative+False\;Positive}$$
(11)$$Accuracy=\frac{True\;Positive+True\;Negative}{Positive+Negative}$$

In the specific classification problem involved in this paper, True Positive refers to a group of people who correctly identify the test, False Negative is a group of people who mistakenly identify the test, and True Negative is meaningless in this classification problem, so the value of True Negative can be regarded as zero.

### 4.3. Performance Analysis of the Method

First of all, we analyzed the accuracy of the method in the experimental environment of this paper and then compared it with other methods. When analyzing the performance of WiGId, we conducted experiments in three different scenarios. When training random forests, the input training data included data from three different environments. When analyzing the performance of the method, the constructed random forest was used to test the real-time data collected in three environments, and the experimental results are shown in Figure 6.

Figure 6 shows the accuracy of the method in three different environments. It can be analyzed that the accuracy of WiGId in the empty hall environment is the highest, the average recognition accuracy is 1% higher than that in the Lab (LOS) environment, and the accuracy change caused by the change of the environment is within an acceptable range, which proves that WiGId has a good ability to resist environmental interference. In the Lab (NLOS) environment, the accuracy of the method is lower. When the number of people identified at one time is 5, the accuracy of the method is 88%, indicating that in the NLOS environment, the external environment affects the accuracy of the method.

The recognition performance of the method is closely related to the scene in which the experiment is carried out, and the multipath effect of the scene is complex, which generally reduces the performance of the method. Therefore, in this study, the experiments were carried out in three different environments, namely, the empty hall, the laboratory, and the partition wall, and the performance of the analysis method is shown in Table 1.

From the above table, it can be concluded that, compared with other classification algorithms for Wi-Fi detection and location, the method proposed in this paper can have higher accuracy in different environments. In the open hall environment, the highest accuracy is 92.36%. In the lab (LOS) environment, the highest accuracy is 91.96%. In the lab (NLOS) environment, the highest accuracy is 91.63%. Because when identifying the group identity, the group is regarded as a whole to extract features, so with the change of the number of people, the detection accuracy fluctuation of the method is also in the range of 5%, which proves that the method proposed in this paper is feasible for group identity recognition.

### 4.4. The Impact of the Depth of Decision Tree on the Accuracy of the Method

We know that the depth of the decision tree will have a direct impact on the classification ability of the decision tree. Generally speaking, too small the depth of the decision tree will reduce the classification ability of the decision tree. In algorithm 1, the splitting stop condition of the decision tree is: (1) the sub data set has been pure; (2) the number of samples in the node is less than the predetermined threshold minimum; (3) the decision tree has reached enough iterations. When the depth of the decision tree is too small, although it saves the time cost, it is often unable to classify effectively. Too much depth of the decision tree will increase the time cost, so the appropriate depth is very important for the decision tree. In order to discuss the influence of the depth of the decision tree on the accuracy of the method, we set different minimum, in turn, to control the depth of the decision tree by setting a minimum. However, the number of decision trees contained in the random forest is fixed. We carried out experiments under the same data set, and the results are as shown in Figure 7.

The graphs of the influence of decision tree depth on the accuracy of the method in the laboratory LOS environment and NLOS environment are shown in Figure 7a,b. It can be seen that the Abscissa represents the maximum depth of the decision tree. With the increase of the depth of the decision tree, the accuracy of classification increases, and the time to build the decision tree also increases. When the maximum depth is from 1 to 10, the accuracy of the method increases linearly and monotonously with the increase of the depth of the decision tree. However, when the maximum depth of the decision tree is greater than 10, the detection accuracy of the method is basically not affected by the maximum depth of the decision tree, so it can be concluded that in the experimental environment of this paper, when the maximum depth of the decision tree is 10 (the construction time is 19.05 s), effective classification results can be obtained without excessive time cost.

### 4.5. The Impact of the Random Forest Size on the Accuracy of the Method

We know that the number of decision trees in the random forest determines the size of the random forest we get. Under the same classification task, different random forests have different classification results. In order to study the influence of the size of the random forest on the classification, control the number of decision trees in the random forest, but do not set the depth of the decision tree, so that each decision tree can grow as much as possible. Except for the different number of decision trees, the rest of each random forest is the same. After building different random forests, test them with the same data set in turn. The results are shown in Figure 8.

### 4.6. The Impact of Equipment Height and Packet Sending Rate on the Performance of Method Recognition

In this section, we have studied the influence of some properties of external hardware on method detection and considered two main factors: (1) the height of the receiver perpendicular to the ground; (2) the packet rate of the sender. The reason for discussing the height of the equipment is based on the Fresnel zone theory, considering that the cutting of different Fresnel zones when the human body is walking has a great influence on the CSI data. Thus, based on the problem to be solved in this paper, the direction, distance, and time of people’s walking were fixed through adjusting the equipment to form different Fresnel zones, and this paper put forward a reference for the placement of equipment to study the problems raised in this paper. In different experiments, the control variable method is adopted; for example, when studying the height of the equipment, the packet sending rate is fixed.

First of all, the height of the transmitter and receiver will affect the collected CSI data, which will ultimately affect the recognition performance of the method. Therefore, we set the height of the transmitter and receiver to remain the same during the grouping experiment, and the experiments were carried out at the height of 0.50 m, 1.00 m, and 1.50 m, respectively. Two reasons are taken into account in setting such a height grouping: (1) the focus of attention varies intuitively with different heights. For an adult of normal height, when the height of the device is 0.50 m, the influence of the lower part of the body on CSI may be greater than that of the upper part of the body, and when the height of the device is 1.50 m, it is on the contrary; (2) considering that if the equipment is to be widely promoted in the future, when placing the equipment, there should be a reference based on the accuracy of the detection method. Then, it can be combined with user-friendliness to get the best way. According to the Fresnel zones theory, we know that different height of the transmitter and receiver will lead to different Fresnel zones, but only part of the Fresnel zones has greater feedback when the human body passes by, and so the cutting areas of the human body walking at different heights are also different; hence, we performed a large number of experiments in three experimental scenarios. Figure 9 shows our experimental results in detail.

Figure 9a–c show the detection accuracy of three methods with different heights of 0.50 m, 1.00 m, and 1.50 m. It can be seen that, generally speaking, the equipment height has an obvious influence on the detection accuracy of the method. The influence of equipment height on the detection accuracy of the method is polarized, that is to say, the detection accuracy of the method will be reduced if the equipment is too high or too low. However, in the middle between low and high, when the height of the equipment is 1.50 m, the detection accuracy of the method is the best.

We speculated that the packet delivery rate might have different effects on personnel identification. In order to verify, under three experimental scenarios, we set the packet delivery rate to 10 different packet rates of 100 p/s, 200 p/s, 300 p/s, 400 p/s, 500 p/s, 600 p/s, 700 p/s, 800 p/s, 900 p/s, and 1000 p/s at the height of the fixed equipment. The results of the experiment are shown in Figure 10.

Figure 10 shows the influence of the packet delivery rate on the detection accuracy of the method. We can see that in the experimental environment of this paper, when the packet delivery rate is less than 500 p/s, the detection accuracy of the method increases with the increase of packet delivery rate. When the packet delivery rate is higher than 800 p/s, with the increase of packet delivery rate, the detection accuracy of the method is basically unchanged, on average, in three different experimental environments. When the packet sending rate exceeds 500 p/s, the detection accuracy of the method fluctuates within 1%, and too high packet sending rate will increase the amount of data that needs to be processed. Therefore, considering the two factors of the data amount to be processed and the accuracy of the method, the packet sending rate should be set between 500 p/s and 1000 p/s.

### 4.7. The Impact of the Number of People on the Performance of Method Recognition

Many methods of perceiving the human body or positioning in the Wi-Fi environment are difficult to achieve high accuracy after the number of people increases because it is difficult to establish the relationship between the characteristics of CSI and the number of people. In the method proposed in this paper, the number of people in the experiment was up to 5, but in view of the situation of many people, we essentially treated multiple people as a whole and did not pay attention to each of them, nor could we identify each of them.

In order to discuss the calculability of the method of treating multiple people as a whole and then matching through the fingerprint library, it is set in the “best” experimental scene, that is, in the empty hall, the equipment height is 1.0 m, and the delivery rate is 1000 p/s. We selected 10 experimenters—the first group only walked one person along the predetermined direction and distance, the second group added one person to the previous group, and so on until we walked through 10 people at once. The data of the experiment are shown in Figure 11.

We can draw from Figure 11 that with the increase in the number of people, the detection accuracy of the method decreases. When the number of people tested is less than 5, the extent of the decline is relatively small; when the number of people tested is more than 8, the range of decline is also relatively small. When the number of people tested is between 5 and 8, the detection accuracy of the method decreases rapidly. We observed that when the number of people tested increases, the signal has more refraction and reflection in the process of propagation than before.

Therefore, it is not easy to establish the relationship between CSI and the number of people, so as the number of people increases, the detection accuracy of the method decreases. But when the number of people tested rises to a certain extent (in the experimental environment of this paper, more than 8 people), the impact of the multi-path effect on the data cannot be distinguished by the method proposed in this paper, so the decline rate slows down and basically remains stable.

## 5. Conclusions

In this paper, in order to solve the problem of identity’s identification, compared with other sensors, considering the advantages of a large number of Wi-Fi devices, no user contact, and not easy to be affected by light, temperature, and humidity in the environment, we choose Wi-Fi devices to identify identities. By extracting CSI information from Wi-Fi signals as features, we establish an identity feature fingerprint database.

The identification method is divided into two stages, the offline stage collects data to establish the fingerprint database, and the online stage uses the fingerprint database matching method to get the recognition results. In order to solve the problem of a large number of data and multi-classification, the random forest is used as the method of classification and matching. The method proposed in this paper is verified by experiments in both the LOS environment and the NLOS environment. The experimental results show that the method has certain accuracy and anti-environmental interference ability in solving the problem of multi-person identification. The best detection accuracy of the method is 92.36%.

To be clear, the method proposed in this paper can reduce the impact of the environment within a certain range, but cannot completely eliminate the impact of the environment.

The deficiency of the method is that when identifying the identity of a multi-person group, multiple people are regarded as a whole, and each person cannot be identified separately in the case of multiple people. We will also look for a model that can distinguish each person in our future work.

## Figures and Tables

**Figure 1 sensors-20-04607-f001:**
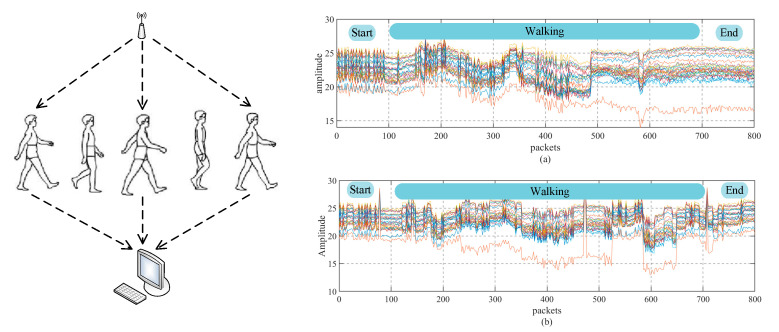
Channel state information (CSI) signal: (**a**) one person walking; (**b**) the other person walking.

**Figure 2 sensors-20-04607-f002:**
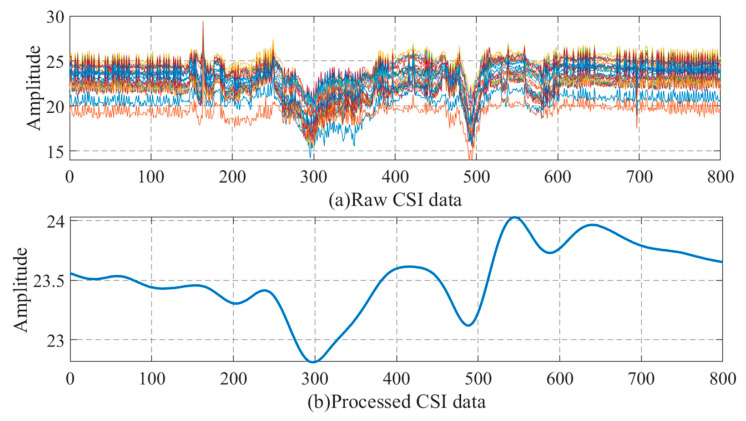
Schematic diagram of CSI data preprocessing. (**a**) Raw CSI data, (**b**) Processed CSI data.

**Figure 3 sensors-20-04607-f003:**
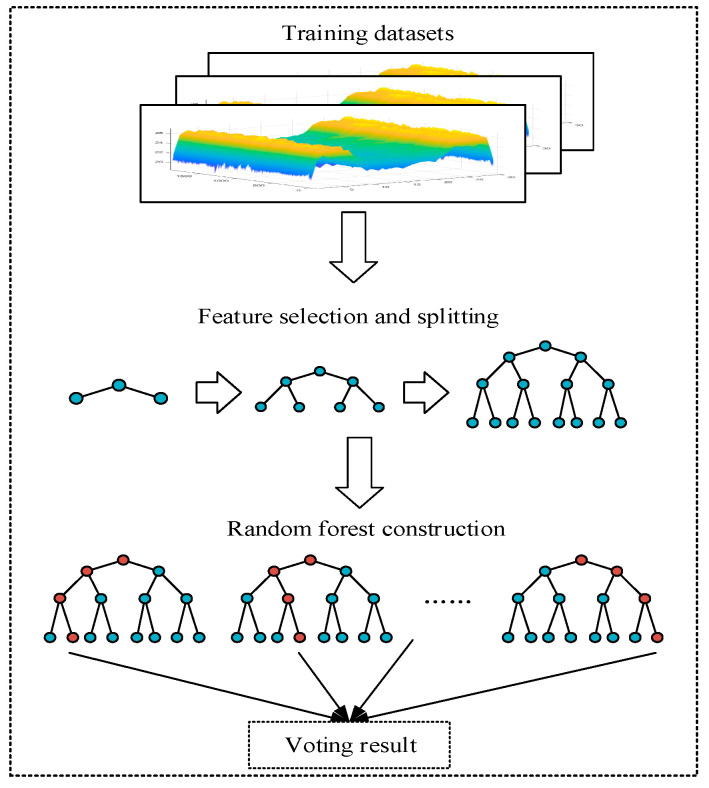
Random forest construction process.

**Figure 4 sensors-20-04607-f004:**
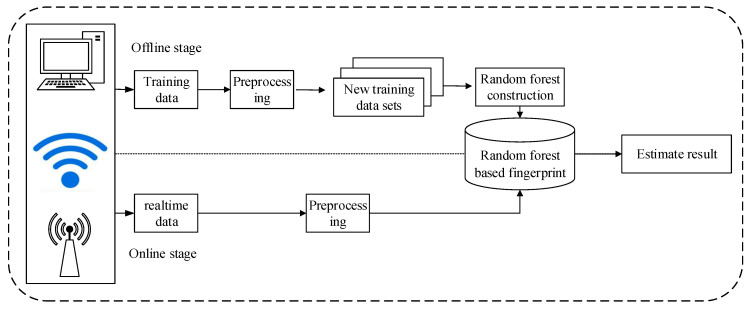
WiGId method flow chart.

**Figure 5 sensors-20-04607-f005:**
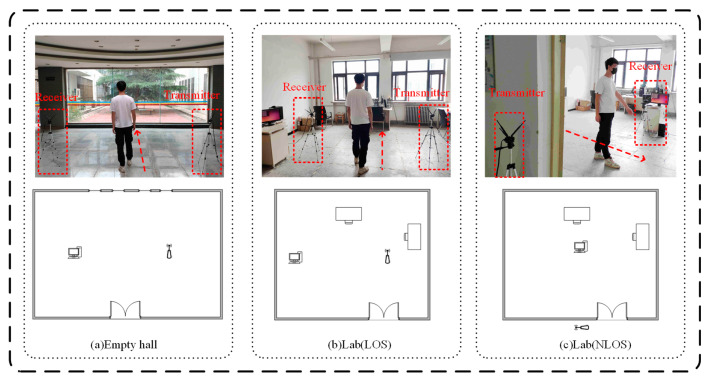
Laboratory and open hall experimental environment. (**a**) Empty hall, (**b**) Lab (LOS), (**c**) Lab (NLOS). LOS, line of sight; NLOS, none line of sight.

**Figure 6 sensors-20-04607-f006:**
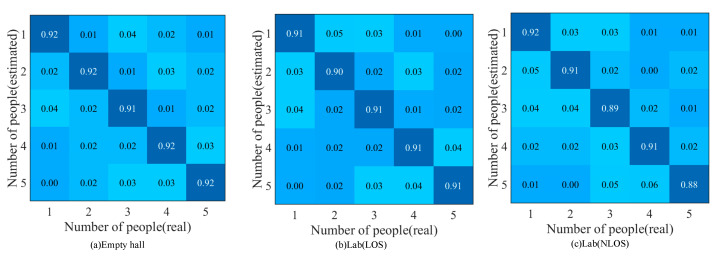
Method performance in three environments. (**a**) Empty hall, (**b**) Lab (LOS), (**c**) Lab (NLOS).

**Figure 7 sensors-20-04607-f007:**
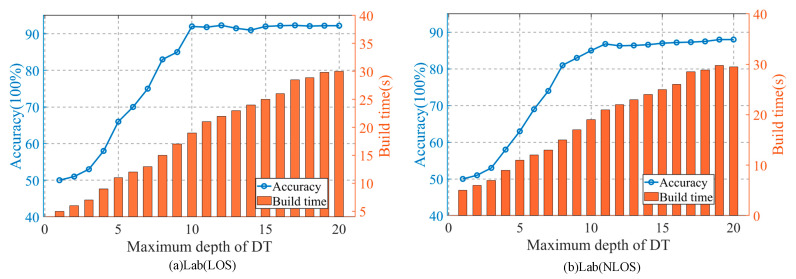
The impact of decision tree depth on the accuracy and construction time of the method. (**a**) Lab (LOS), (**b**) Lab (NLOS).

**Figure 8 sensors-20-04607-f008:**
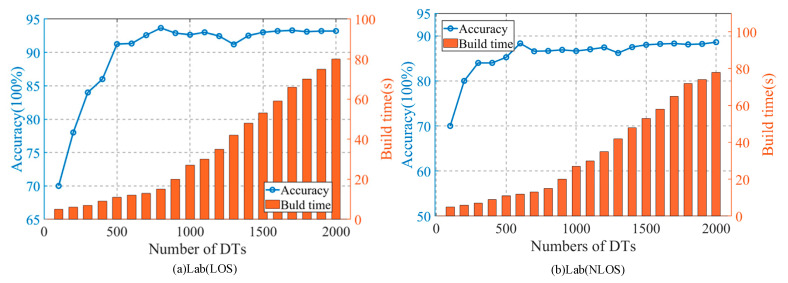
The impact of the size of the random forest on the accuracy of the method and construction time. (**a**) Lab (LOS), (**b**) Lab (NLOS).

**Figure 9 sensors-20-04607-f009:**
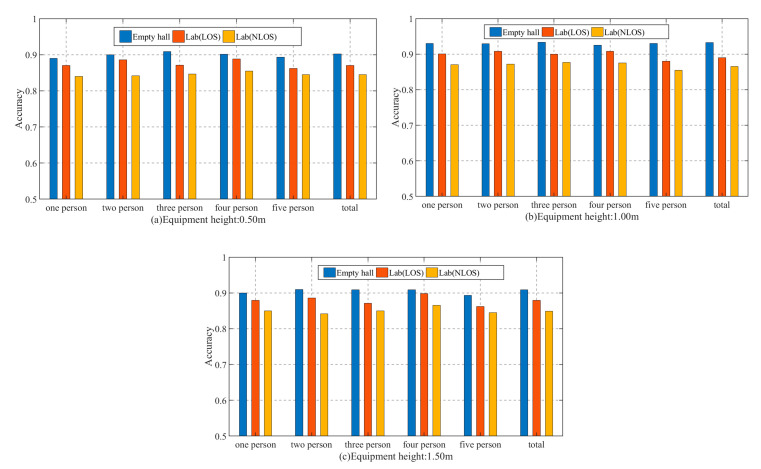
The impact of equipment height on accuracy in different environments. (**a**) Equipment heights: 0.50 m, (**b**) Equipment heights: 1.00 m, (**c**) Equipment heights: 1.50 m.

**Figure 10 sensors-20-04607-f010:**
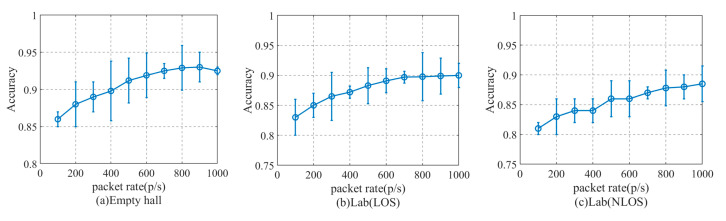
The impact of different packet delivery rates on the accuracy of the method. (**a**) Empty hall, (**b**) Lab (LOS), (**c**) Lab (NLOS).

**Figure 11 sensors-20-04607-f011:**
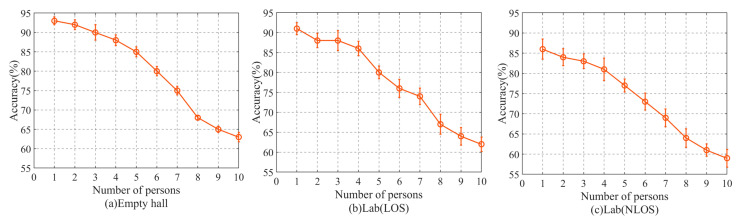
The influence of the number of people on the accuracy of the method. (**a**) Empty hall, (**b**) Lab (LOS), (**c**) Lab (NLOS).

**Table 1 sensors-20-04607-t001:** Performance comparison table of WIGID and other algorithms.

Different Experimental Scenarios and Methods		Accuracy of the Method (%)				
		One-Person	Two-Person	Three-Person	Four-Person	Five-Person
Empty hall	SVM (RBF)	80.25	81.92	83.78	82.25	80.96
	WFID	85.20	83.69	84.32	81.36	80.12
	WiGId	92.05	92.08	91.03	91.95	92.36
Lab (LOS)	SVM (RBF)	78.95	79.65	76.33	78.02	76.94
	WFID	80.16	84.01	82.21	85.69	84.78
	WiGId	91.96	89.94	89.61	90.76	91.15
Lab (NLOS)	SVM (RBF)	75.32	72.36	70.66	73.65	71.32
	WFID	80.08	78.17	78.33	75.25	72.56
	WiGId	91.63	89.60	89.33	89.57	88.21

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
