# Peer review of "WiGId: Indoor Group Identification with CSI-Based Random Forest"

_sensors, 2020, doi:10.3390/s20164607_

Round 1
Reviewer 1 Report
The article proposed an algorithm to identify small-scale (2-5) group personnel at one time using features obtained from channel state information (CSI), using a random forest classifier. The performance of the proposed approach is evaluated using both LOS and NLOS APs using two main configurations (adding a concrete wall).
Related to the paragraph "The CSI is more reliable than the RSSIs that change frequently depending on the time and environment", authors must discuss arguments in favor of this afirmation by including some references (maybe the following paper could be useful: https://doi.org/10.1145/2543581.2543592 )
There are some mistakes in the abstract:
1) "When extracting channel state information ((channel state information, CSI) features,...."
2) The terms LOS (line of sight (LOS) ?) and NLOS (Non-line-of-sight (NLOS) ?) must be defined (currently, there is not a definition in the paper)
There are some minor details about missing spaces in lines 42, 76, 80, 83, 86, 94, 124, 364. I suggest to check the text carefully in the next version of this manuscript.
The paragraph line spacing varies in some sections, so the authors must review carefully for a future submission (this is especially visible In section 4.6)
The related work section should be extended in order to include some relevant papers. The following related relevant papers should be cited, including some coments about the advantages and disadvantages of the proposed approach:
https://doi.org/10.3390/s18113987
https://ieeexplore.ieee.org/abstract/document/8830368
https://ieeexplore.ieee.org/abstract/document/8767421
https://doi.org/10.1155/2018/1253752
Reviewer 2 Report
The discussion of the state of the art and the list of references should be expanded.
The set of features and its extraction is not well described. I think mathematical Equations may help to understand how features are computed. Authors should explain if the set of features are computed on a single realizations of the channel matrix H or, in contrast, features are computed over an observation window of W realizations as in:
M. De Sanctis et al. “WIBECAM: Device Free Human Activity Recognition Through WiFi Beacon-Enabled Camera”, Proceedings of the 2nd workshop on Workshop on Physical Analytics (WPA’15), May 22, 2015.
The problem of environment-dependence of the results should be discussed at least in the Introduction Section, also referring to past works such as:
Wenjun Jiang et al, “Towards Environment Independent Device Free Human Activity Recognition”, MobiSys 2018.
Simone Di Domenico et al, “Exploring Training Options for RF Sensing Using CSI”, IEEE Communications Magazine, vol. 56, no. 5, May 2018.
Reviewer 3 Report
The paper proposes a mechanism to identify the number of people by using WiFi with a improved random forest algorithm.
The paper shows very promising results that are either compared with other algorithms. However, there are some important flaws in the document from both, formal point of view and content point of view.
From the formal point of view there are some text from the original template, and from the content point of view there are some elements that, from my point of view, need an improved explanation in order to better understand what authors have done.
The paper needs a review of English, either.
Please, find attached some comments and suggestions. Hope authors find them useful to improve the paper.

Round 2
Reviewer 2 Report
This version is fine
Author Response
Thank you.
Reviewer 3 Report
The paper dealt with the previous comments and, from my point of view, it gained in clarity. Please, find in the attached document the proposed elements that I think can help to improve the paper.
